# Deep Chemical and Physico-Chemical Characterization of Antifungal Industrial Chitosans—Biocontrol Applications

**DOI:** 10.3390/molecules28030966

**Published:** 2023-01-18

**Authors:** Gaël Huet, Yunhui Wang, Christian Gardrat, Daphnée Brulé, Amélie Vax, Cédric Le Coz, Frédérique Pichavant, Silvère Bonnet, Benoit Poinssot, Véronique Coma

**Affiliations:** 1Laboratoire de Chimie des Polymères Organiques, Université de Bordeaux, CNRS, Bordeaux INP, UMR 5629, 16 Avenue Pey-Berland, F-33600 Pessac, France; 2Agroécologie, CNRS, INRAE, Institut Agro, University Bourgogne, University Bourgogne Franche-Comté, F-21000 Dijon, France; 3Elicityl SA, 746 Avenue Ambroise Croizat, F-38920 Crolles, France

**Keywords:** chitosan characterization, SEC-MALS, NMR, XPS, elemental analysis, TGA, mass spectrometry, antifungal properties, bioactivity

## Abstract

Five different chitosan samples (CHI-1 to CHI-5) from crustacean shells with high deacetylation degrees (>93%) have been deeply characterized from a chemical and physicochemical point of view in order to better understand the impact of some parameters on the bioactivity against two pathogens frequently encountered in vineyards, *Plasmopara viticola* and *Botrytis cinerea*. All the samples were analyzed by SEC-MALS, ^1^H-NMR, elemental analysis, XPS, FTIR, mass spectrometry, pyrolysis, and TGA and their antioxidant activities were measured (DPPH method). Molecular weights were in the order: CHI-4 and CHI-5 (MW >50 kDa) > CHI-3 > CHI-2 and CHI-1 (MW < 20 kDa). CHI-1, CHI-2 and CHI-3 are under their hydrochloride form, CHI-4 and CHI-5 are under their NH_2_ form, and CHI-3 contains a high amount of a chitosan calcium complex. CHI-2 and CHI-3 showed higher scavenging activity than others. The bioactivity against *B. cinerea* was molecular weight dependent with an IC50 for CHI-1 = CHI-2 (13 mg/L) ≤ CHI-3 (17 mg/L) < CHI-4 (75 mg/L) < CHI-5 (152 mg/L). The bioactivity on *P. viticola* zoospores was important, even at a very low concentration for all chitosans (no moving spores between 1 and 0.01 g/L). These results show that even at low concentrations and under hydrochloride form, chitosan could be a good alternative to pesticides.

## 1. Introduction

The 21st century has its fair share of challenges regarding environmental issues. One of them is the intensive use of chemicals and pesticides in the agricultural world. As we all know, those products can lead to some issues, such as soil pollution or health concerns [1,2]. However, crop production is heavily dependent on such substances as our agricultural system is currently very intensive. Intensive farming is prone to diseases that are more likely to spread extremely fast if the plants are not protected. Thus, the use of harmful pesticides for the environment and health was considered a necessity to be able to provide enough commodities. A perfect example of the intensive use of pesticides is observable in wine production. Grapevine (*Vitis vinifera*) is subject to numerous diseases like downy mildew coming from *Plasmopara viticola* or other fungi like *Botrytis cinerea* that can alter wine quality and production [3,4,5,6]. Both those pathogens can be really problematic for vineyards and even destroy all the harvest if not controlled. It became a real challenge for wine producers to protect their vineyards with respectful products for the environment and for their own health.

In this context, the development of effective and eco-friendly pesticides is becoming a challenge. Chitosan, an active and antimicrobial biobased polymer, has a strong potential in terms of biopesticides with low environmental impact. It consists of β-(1-4) linkage of units of D-glucosamine (D-units) and N-acetyl-D-glucosamine (A-units) with a variable degree of acetylation (Figure 1). It is produced from the deacetylation of chitin, which is mainly found in crustacean shells, insect cuticles and fungal cell walls. In this study, chitosans from crustaceans have been chosen, as chitosans obtained from fungal cell walls can have a lot of β-glucans that reduce their intrinsic bioactivity [7]. Chitosan is known to have antifungal and antibacterial properties against several pathogen species and strains and was able to reduce infection on different plants organs (leaf, flower, fruits) and on different crops (wheat, potatoes, tomatoes, grapevine…) [8,9,10,11,12,13,14]. Moreover, chitosan also possesses elicitor properties on several plants and thus has a double effect on plants with direct and indirect protection [15,16,17,18]. The antimicrobial activity of chitosan depends on its physicochemical properties as chitosans with molecular weight higher than 5 kDa will not be able to go through the membrane and will act as a chelator of essential metals, prevent nutrients from being taken up by the microorganism and can interact with the fungal cell wall [19,20,21]. Chitosan oligomers with a size lower than 5 kDa also can go through the membrane and affect DNA/RNA protein synthesis and even mitochondrial activity [19,21,22,23]. Besides these properties, chitosan is biodegradable, which is also an important factor in the design of an eco-friendly biopesticide [24,25]. Although the chitosan effect on microorganisms and plants starts to be well known, the relationships between the bioactivity of chitosan and some structural and physicochemical characteristics such as the degree of deacetylation, the molecular weight, the polymerization degree (DP), the pH of the chitosan solution and the solubilization process are not always clear and currently being discussed [26]. This is a consequence of the fact that relatively few scientific publications based on chitosan study in depth these chemical and physicochemical parameters, making it difficult to compare scientific results.

Moreover, producing chitosan on a large scale can be challenging for industrial companies, and the quality of chitosan can suffer during the process, which could alter its properties. External particles can interfere with experiments and characterizations. This is why a large part of this study will be about chitosan purity and how to determine the composition of chitosans. Five different samples (CHI-1 to CHI-5) with high deacetylation degrees (DAs) have been studied to better understand the impact of the molecular weight and other chemical and physicochemical parameters on the bioactivity against *P. viticola* and *B. cinerea*.

## 2. Results

### 2.1. Visual Observation of Chitosan

A color gradient can be seen from brown to white from CHI-1 to CHI-5 (Figure 2). Also, some white particles can be observed in CHI-2 and CHI-3.

### 2.2. Bioactive and Antioxidant Properties 

#### Evaluation of the Antifungal Activity on B. cinerea and P. viticola

To further characterize CHI-1 to CHI-5, we investigated their antifungal activity on the necrotrophic fungus *B. cinerea* and the biotrophic oomycete *P. viticola*. We first tested their impact on *B. cinerea*’s mycelial growth in vitro. The growth of *B. cinerea* was inhibited by all the chitosans at low concentrations, but some differences have been observed. CHI-1 and CHI-2 seem to be the most fungitoxic, with a half maximal inhibitory concentration (IC50) of 13 mg/L. CHI-3 and CHI-4 were more variable and showed intermediate antifungal activity with an IC50 of 17 or 75 mg/L, respectively. Compared to the others, CHI-5 is clearly less toxic on *B. cinerea* with an IC50 of 152 mg/L and highly variable (Table 1 and Appendix A).

The toxicity of CHI-1 to CHI-5 was then assessed on *P. viticola* by counting the number of moving zoospores after chitosan or water treatment. All the different chitosans are toxic at very low concentrations to *P. viticola* zoospores. Sporangia treated with CHI-1 to CHI-5 at concentrations from 1 to 0.01 g/L did not release any moving zoospore (Figure 3). At 0.005 g/L, only CHI-4 is no longer fungitoxic. At 0.001 g/L, there are as many moving zoospores as in the control, except for CHI-2, which is the most toxic. These results suggest a strong direct biocide effect of the chitosans showing different bioactivities between CHI-1 to CHI-5.

### 2.3. Antioxidant Activity

2,2-Diphenyl-1-picrylhydrazyl (DPPH) scavenging method was used to evaluate the antioxidant activity of the different chitosan samples [26,27,28]. DPPH is a stable radical that appears deep purple in an ethanolic solution. In the presence of antioxidant molecules, a reaction occurs, and there is a decrease in the optical density at 517 nm leading to a color change of the solution from purple to pale yellow. Scavenging activities were calculated according to Equation (1) in materials and methods. Results are reported in Table 2. It can be seen that, under the experimental conditions used, CHI-1 and CHI-5 do not have an important scavenging activity, CHI-2 and CHI-3 showed a higher one, and CHI-4 showed an intermediate one. Nevertheless, it should be noted that ascorbic acid, a well-known antioxidant compound, gave a scavenging activity of about 96% in the same experimental conditions but at a ten times lower concentration than for the chitosan samples. Obviously, the chitosan samples were not a very effective scavenger for DPPH radicals. 

### 2.4. Characterization of Chitosans

#### 2.4.1. Deacetylation Degree from ^1^H-NMR Spectrometry

As expected, all chitosan samples have a high deacetylation degree, between 93% and 98%. DAs have been determined using ^1^H-NMR (Table 3 and Appendix A). No impurities, such as proteins, were found.

#### 2.4.2. Molecular Weight Determination

##### Molecular Weight from SEC-MALS Methods

In this study, two widely used SEC MALS methods have been performed on different molecular weight chitosans. One method is based on the sodium acetate buffer (NaAc; pH 4.5), while the other one is based on the ammonium acetate buffer (NH_4_Ac; pH 4.5). Buffers with a pH of around 4.5 are known to prevent the formation of chitosan aggregates due to their proper ionic strength. Both those buffers have been widely used for the determination of the molecular weight of chitosans due to their ability to solubilize samples [29,30,31,32,33,34]. 

All results are shown in Table 4, and chromatograms are available in Appendix A. The solubility of chitosan in the buffer is an important factor as samples are filtered before analysis. Poor solubility could lead to a loss during filtration and, thus, a biased analysis of the molecular weight. This is why samples have been solubilized overnight. For the lower molecular weight chitosans, the solubility problem is less important as they were totally soluble in both buffers. However, sometimes the results are not reliable as the same chitosan did not get the same molecular weight in both buffers. CHI-2 is a prime example where both buffers gave different results in Mn and Mw with 8.0 kDa and 12.1 kDa, respectively, in an NH_4_Ac buffer against 14.4 kDa and 17.5 kDa in the NaAc buffer, and similar differences can be observed on all samples. The polydispersity of different samples showed similar results for both methods. Nevertheless, differences in polydispersity are still observable in CHI-3 and CHI-4.

##### Polymerization Degree from ^1^H-NMR Method

^1^H-NMR is generally used to determine the size of oligosaccharides using Formula (3) in materials and methods. The results are presented in Table 5, and the NMR spectra are in Appendix A. Using this method, CHI-1 and CHI-2 showed a DP 15 and DP 44, respectively. However, Equation (3) does not allow us to determine the molecular weight of CHI-3 as results were much higher than 50 with a DP up to 113. Above DP 50, the integration becomes less accurate, and the margin of error can become quite high.

To be sure that no hydrolysis of the sample occurred during the solubilization or the NMR analysis, tests were performed on chitosan with an intermediary DP (DP = 50) at various contact times and no reduction in DP was observed, and no hydroxymethylfurfural (HMF) [34] was detected (Appendix A). 

#### 2.4.3. Elemental Analysis

The elemental analysis of CHN and O is presented in Table 6. As the amount of water in the chitosan chains seemed to be practically the same for all the chitosans (TGA results), hydrogen and oxygen results are included in Table 6. It can be observed that the sum of percentages of all the different elements analyzed (C, H, N, O) was largely under 100% for CHI-1, CHI-2 and CHI-3, whereas it was closer for CHI-4 and CHI-5. 

#### 2.4.4. X-Ray Photoelectron Spectrometry 

XPS allowed us to determine the surface chemical composition of the different samples. All results are presented in Table 7 and in Appendix A. As the amount of water present in the different chitosans was almost the same (TGA analysis), the hydrogen values from the elemental analysis have been added to the calculation to obtain a better approach to the surface chemical composition. It is interesting to note that the sum of hetero atoms was similar to the amount obtained by elemental analysis (Table 6). Regarding the chlorine atoms, they were present on the surface of CHI-1, CHI-2 and CHI-3 in a larger amount than for CHI-4 and CHI-5. The presence of calcium atoms was surprising and particularly on CHI-3’s surface. The high-resolution spectra of Ca 2p showed peaks of Ca 2p_1/2_ at 349.2 eV and Ca 2p_3/2_ at 345.6 eV. This is compatible with the presence in all the samples with more or less quantities of calcium-chitosan complexes, as described by Wei et al. (2021) [35].

#### 2.4.5. Infra-Red Spectroscopy

FTIR spectroscopy is a versatile tool in the characterization of chitosan. It provides qualitative information on chitosan purity. As observed in Figure 4, high molecular weight chitosans, CHI-4 and CHI-5, have shown NH_2_ form with the stretching C=O of amide I at 1655 cm^−1^ and bending vibration of NH_2_ at 1590 cm^−1^ [36,37,38]. However, for CHI-1, CHI-2 and CHI-3, only one band appeared at 1628 cm^−1^. That could be related to metal complexation that might occur during the production process [39]. Chitosan CHI-1 and CHI-2 presented a planned-bending vibration of NH_3_^+^ at 1515 cm^−1^ [36] that could be related to the protonation of chitosan. Vibrations at 1423 cm^−1^ and 1470 cm^−1^ are attributed to the coupling of C-H axial stretching and N-H angular deformation [38]. The band at 1423 cm^−1^ present in CHI-3 is particularly important and could come from impurities.

#### 2.4.6. Mass Spectrometry

Mass spectrometry can be used to characterize chitooligosaccharides (COS), but studies have been limited to species that only contain a few residues with a low degree of polymerization. Polymers of chitosan have not been observed yet using mass spectrometry because no multiply charged ions are formed using electrospray or MALDI techniques, indicating that these soft methods were inefficient for the complete analysis of these biopolymers. Moreover, the experimental conditions seemed to be very important in determining the type of COS present in chitosans by mass spectrometry [40,41,42].

In the electrospray (+) mass spectra of CHI-1 and CHI-3, several ions were detected, which corresponded to protonated ions accompanied by their dehydrated ions, the latter presenting higher relative intensities (Appendix A). So, ions at *m*/*z* 162/180, 323/341, 484/502, 645/663, 806/824, 967/985, 1128/1146 and 1289/1307 were easily identified. They corresponded to completely deacetylated structures D1 to D8; no acetylated peaks appeared in these spectra. As the mass to charge increased, intensities of peaks decreased as it is well known for mass spectra of chitosan oligosaccharides. As dissociation of chitosan and COSs can occur in the electrospray source [43], it is interesting to use another mass technique like MALDI TOF to determine the structure of COS. In the MALDI (+) mass spectra of CHI-1 and CHI-3, cationized ions (Na^+^) were detected using the reflectron mode (Appendix A). The most abundant were identified as deacetylated structures from *m*/*z* 524 (D3) to *m*/*z* 1490 (D8) accompanied by very smaller dehydrated ions. Interestingly, in these spectra, some deacetylated ions with low abundances appeared at *m*/*z* 567, 727, 888 and 1049, corresponding to D2A, D3A, D4A and D5A, respectively. As for electrospray, as the mass to charge increased, intensities of peaks decreased, and higher *m*/*z* cannot be observed in these cases. The reflectron mode did not give any spectrum for chitosans CHI-2, CHI-3 and CHI-5, but when using the linear mode (Appendix A), several well-separated clusters appeared, which could correspond to COS with DP 3 to 9. Moreover, the spectrum of chitosan CHI-1 registered in the linear mode indicated the presence of COS from D5 (*m*/*z* 847) to D15 (*m*/*z* 2459). Those results confirmed that all samples are highly deacetylated, as the NMR results suggest.

#### 2.4.7. Pyrogram

Isothermal pyrolysis of chitosans CHI-1, CHI-2, CHI-3, CHI-4 and CHI-5 was undertaken at 500 °C. The evolved compounds were separated by gas chromatography and identified by mass spectrometry. Mass spectra were obtained by electron ionization, and compound identification was made by comparison with a mass spectra library. In total ion chromatograms (TICs), a large number of peaks corresponding to low-abundance compounds were poorly resolved, and it was difficult to identify them with certainty using only the library. Moreover, when there is a large quantity of very volatile products released, the relative abundance of the other ones is reduced, and the comparisons between the pyrolysis products of the different chitosans were very difficult. All TICs are presented in Appendix A, and the best proposal identification is in Appendix A with references [38,44,45,46,47,48,49,50,51,52,53,54].

The comparison of the TICs Indicated different behaviors of the chitosans. For chitosans CHI-1 and CHI-2, more amounts of volatile products (carbon dioxide and hydrogen chloride) were evolved during pyrolysis than in the other ones. Surprisingly, the TIC of chitosan CHI-3 presented practically no volatile compounds. Moreover, in the TICs of chitosans CHI-1, CHI-2 and CHI-3, a curious, very large asymmetric peak (t_R_ between about 13.5 and 18.7 min (Appendix A)) appeared, which was not identified by the library. The only data obtained by selected ion monitoring (*m*/*z* 36) indicated the presence of hydrogen chloride. So, this peak was likely due to a compound in its hydrochloride form. The TICs of chitosan CHI-4 and CHI-5 were very similar, and they showed the presence of carbon dioxide but not the formation of hydrogen chloride. In the pyrograms, already described compounds based on furan, pyrazine and pyrrole structures formed during chitosan pyrolysis were identified.

#### 2.4.8. Thermogravimetric Analysis (TGA)

TGA is useful for giving information such as the water content and the thermal resistance of the substance. The results are listed in Table 8, and the thermograms and the derivatives’ dTGAs are presented in Figure 5 and Appendix A, respectively.

As expected, the thermal resistance decreased from about 303 °C to 201 °C for CHI-5 to CHI-1 due to the decreasing molecular weight. The maximum value of the derivate thermogravimetric curve (DTGmax) can be an important factor as it is suspected that treatments carried out on chitosan can modify the conformation of fibers and make them more or less resistant to thermal degradation. 

Water content and ash percentage can be determined by TGA and allow quantifying the general purity of the chitosan [55]. CHI-2 has a high ash percentage of 8.3%, but CHI-3 and CHI-4 also obtain important ash percentages of 4.4% and 5.2%, respectively. CHI-1 has a very low ash percentage of only 1.2%. 

## 3. Discussion

### 3.1. Deacetylation Degree

All techniques of DA determination have advantages and drawbacks, but NMR is considered the most reliable of them as it is still possible to determine the DA with inorganic impurities, and it is quick and easy. All samples have high deacetylation degrees (superior to 93%), which was confirmed by mass spectrometry. Although this DA determination method is performed in a mixture of D_2_O: DCl, there is no hydrolysis occurring as all samples were at the same DA, no matter the temperature or time of solubilization. 

### 3.2. Molecular Weight

The determination of the molecular weight of chitosans is complicated because of the generally poor solubility of chitosans and the poor reliability of the methods used. The solubility of chitosan depends on its DA, its molecular weight, its general purity and the solvent used [56]. Several methods exist to determine the molecular weight of chitosan as the viscosity measurement [57], but SEC-MALS is usually considered the most direct method and is frequently used. However, depending on the process, chitosan is known to form aggregates and can possibly give biased results. The formation of aggregates is dependent on several factors as deacetylation degree [58], ionic strength, pH and degree of polymerization [31,47].

Sometimes, results can vary from one method to another, as can be seen with CHI-2, where Mn varies from 8 to 14 kDa, using, respectively, the NH_4_Ac or NaAc buffer-based methods (Table 4). Those differences in molecular weight, as well as the higher value obtained compared with the NMR results, could be due to aggregation, but this is generally not expected with such low molecular weight in the case of CHI-1 and CHI-2. The concentration of salt is a major parameter in the formation of aggregates, and low molecular weight chitosans are more prone to form aggregates [32,34,58]. Also, it is possible that the presence of chlorine, calcium or silicium and the fact that some chitosans are protonated are factors to take into account for the solubility and the formation of aggregates into both buffers. A lower buffer salt concentration could prevent the formation of aggregates for low molecular weight chitosan [32]. It is difficult to choose between the two buffers as their results are not exactly the same. However, it can be seen that results have a similar trend when the molecular weight increases from CHI-1 to CHI-5, and this is for both methods with a relatively similar scale. 

Smaller molecular weight samples (CHI-1, CHI-2 and CHI-3) present a lower polydispersity than the higher ones (CHI-4 and CHI-5). This could be due to the purification step carried out to recover smaller oligosaccharides. Because of the high length of higher molecular weight chitosan, it is more difficult to obtain a polydispersity close to 1. Except for a few exceptions like CHI-3 and CHI-4, all samples showed similar polydispersity for both methods. An important difference could have suggested an insolubility issue and, thus, a filtration problem. Although, a high polydispersity difference could also mean an aggregation of chitosan. As the difference between both methods is not significant, it does not provide supplementary information to compare Mn and Mw.

Molecular weight determination by NMR is useful to get around the problem of aggregates from the SEC-MALS analysis. It gives more precise DP determination for oligomers (DP < 50) [59,60,61]. For lower molecular weight chitosans, ^1^H-NMR should be preferred, especially since no hydrolysis occurs during solubilization. The NMR method cannot be used for higher molecular weight chitosans such as CHI-3, CHI-4 and CHI-5 because, above DP50, the measurement becomes less and less precise. 

TGA results were in line with the chitosan size results, showing an increasing DTGmax from CHI-1 to CHI-5. 

To conclude, CHI-1 and CHI-2 are low molecular weight chitosans, CHI-3 is a medium molecular weight chitosan and CHI-4 as CHI-5 can be considered as high molecular weight chitosans. 

### 3.3. Chitosan Composition 

The presence of unusual contaminants in the different chitosans has been detected through several analyses. They are found in higher proportions in the smaller CHI-1, CHI-2 and CHI-3 (Table 7). TGA (Table 8) indicated important quantities of ashes (>5%) in CHI-2, CHI-3 and CHI-4. Generally, after chitin extraction from crustacean shells, the ash percentage should not exceed 5% [55,62]. XPS analyses (Table 7) showed a relatively high chlorine amount (>5%) in CHI-1, CHI-2, CHI-3 and CHI-3, a high calcium amount (~7%). As a confirmation of chlorine, pyrolysis-GC/MS (Appendix A) showed the presence of hydrogen chloride in CHI-1, CHI-2 and CHI-3. From the infrared characterization of chitosans (Figure 4), the shifts of the C=O bond observed in the spectra of CHI-1 and CHI-2 are in accordance with the presence of contaminants in the samples. A band corresponding to the formation of NH_3_^+^ appears only in the spectra of CHI-1, CHI-2 and CHI-3. Moreover, in the spectrum of CHI-3, a new band appears, which is compatible with the presence of a metallic complex.

Based on all these results obtained from the different analytical methods used, it can be assumed that:(i)CHI-1, CHI-2 and CHI-3 are under their hydrochloride form;(ii)CHI-4 and CHI-5 are under their NH_2_ form;(iii)CHI-3 contains a high amount of chitosan calcium complex.

The hydrochloride forms of CHI-1 and CHI-2 could be due to the hydrolysis and purification process, as hydrochloric acid is often used to hydrolyze chitosan or chitin [63]. Calcium could come from an incomplete chitin extraction as calcium carbonate enters into the composition of crustacean shells.

### 3.4. Bioactivity and Antioxidant Activity

It is well known in the literature that chitosans with high DA are more bioactive due to their higher charge density as more amine functions are free to be protonated [64]. Positively charged chitosan will react more effectively with the membranes of microorganisms [21]. Chitosans have been shown to have better bioactivity if highly deacetylated [65]. Thus, it was important for all samples to have a high DA as it would provide better antimicrobial properties and it would be more effective in a biopesticide formulation [21,26,64].

The composition and the forms of chitosans seem to play a part in the antioxidant activity, where highly contaminated samples showed to have better scavenging effects at a concentration of 1.6 mg/mL (Table 2). Different chitosan forms and complexes could play a part in the bioactivity as it could reduce it due to lower chitosan concentration or improve it. This, combined with their lower molecular weight that already enhances their antifungal properties, could also be one of the explanations for their higher performances. Also, positively charged chitosans are known to interact with microorganism membranes and show a bioactivity effect [20,22]. 

Antifungal properties of chitosan on *B. cinerea* seem to be molecular weight dependent. SEC-MALS data indicate that the molecular size chain increased from CHI-1 to CHI-5. The growth inhibition assays of *B. cinerea* after chitosan treatment suggest that CHI-1 and CHI-2 (IC50 = 13 mg/L) possess the most fungicidal activity. These results are in agreement with the literature confirming that the higher antimicrobial activity is described to be heavily dependent on the molecular weight and the deacetylation degree of chitosan [66,67,68]. Chitosan, with a molecular weight lower than 5 kDa, binds with DNA and disrupts protein synthesis or even alters the proper functioning of mitochondria [19,20,21,22,23]. On the contrary, CHI-3, CHI-4 and CHI-5 seem to be less fungitoxic as the molecular size chain increases. High molecular weight chitosan has been reported to show extracellular antimicrobial activity as it could chelate nutrients or ions and interact with the cell wall leading to the death of the microbes. The interaction with the cell wall could be stronger if the membrane gets high unsaturated fatty acid content which gives greater fluidity to the membrane and more negative charges on the cell wall [69,70]. The more the molecular weight increases, the less chitosan is soluble in water which could also explain a lower antifungal activity. To perform these experiments, we used lactic acid to solubilize high molecular weight chitosans, a solvent that has no direct effects on both pathogens.

However, the results of this study also indicated that this claim is not totally similar to *P. viticola*. All the chitosans were very effective at very low concentrations (5 mg/L), even with the one with the higher molecular weight, CHI-5. CHI-1 and CHI-2 are still very fungicidal, and CHI-2 is significantly the most toxic from 1 mg/L (not significant for CHI-1). Nevertheless, some white particles visible to the naked eye were observed on the lower molecular weight chitosans CHI-2 and CHI-3 (Figure 2), which could also play a part in their stronger bioactivity. These samples contain high chlorine and calcium levels, which makes difficult the interpretation of scavenging effects. In those conditions, CHI-2 and CHI-3 showed great scavenging activity, and CHI-1 and CHI-5 showed a similar scavenging effect of pure chitosan as chitosan is not a really strong antioxidant polymer without modification [14,26,55,71]. Taken together, these results suggest that working on unpurified or inaccurately purified products may be a challenge to establish a direct correlation between bioactivity, antioxidant properties and physicochemical properties.

## 4. Materials and Methods 

### 4.1. Materials

Chitosans from crustacean shells with high DA and different molecular weights were provided by Elicityl (Crolles, France). Five different batches of chitosan were studied, CHI-1, CHI-2, CHI-3, CHI-4 and CHI-5, and were stored at room temperature and sheltered away from the sun. 

All materials were used without any further purification. All deuterated solvents were provided by Eurisotop (Saint-Aubin, France). Acetic acid (96%) was supplied by Fisher chemical (Waltham, MA, USA). Sodium acetate, ammonium acetate and ascorbic acid were supplied by Alfa Aesar (Haverhill, MA, USA). DPPH was supplied by TCI (Tokyo, Japan). Deionized water, with a resistivity of 18.3 MΩ cm, was used for all the experiments.

Grapevine downy mildew (*P. viticola*) was routinely maintained on *Vitis vinifera cv. Marselan* plants as previously described [72].

The BMM strain of *B. cinerea* used [73] was grown on Petri dishes containing V8 medium ½ diluted, KH_2_PO_4_ 5 g/L, agar 30 g/L, pH 6.0 for two weeks in the dark (22 °C). Conidia were collected with water, filtered to remove mycelia, counted and kept at 4 °C prior to infection assays.

### 4.2. Methods

#### 4.2.1. Antioxidant Activity: 2,2-diphenyl-1-picrylhydrazyl Radical Assay (DPPH)

20 mg of chitosan or ascorbic acid (positive control) have been solubilized into 2 mL of an aqueous solution of a 1% (v/v) acetic acid solution and stirred for 24 h. An ethanol solution with DPPH (120 µmol/L) was prepared on the day of the experiment. Dilutions were prepared with 2 mL of the DPPH solution, and dilutions were prepared with acetic acid to reach a concentration of 1.6 mg/mL for chitosan or 0.16 mg/mL for ascorbic acid. The solutions were stirred with a vortex and kept away from the light for 30 min. Then UV scans were performed, and the adsorption was measured at 517 nm using an Agilent Cary Series UV-Vis spectrophotometer. Blanks have been measured using pure ethanol instead of the DPPH solution. All experiments have been repeated 3 times. The control solution was done with a 1% acetic acid solution. The scavenging effect was then determined using Formula (1) [14]:(1)$$\mathrm{Scavenging}\;\mathrm{activity}=\left(1-\frac{A517\;Sample-A517\;Blank}{A517\;Control-A517\;Blank}\right)\times 100$$
where *A*517_Sample_ is the absorption of the sample at 517 nm, *A*517_Control_ is the absorption of the control, and *A*517_Blank_ is the absorption of the blank.

#### 4.2.2. NMR

^1^H-NMR analyses were performed at room temperature using Liquid-state 400 MHz NMR spectrometer (Bruker ADVANCE I, Billerica, MA, USA) using a 5 mm Bruker multinuclear z-gradient direct probe. ^1^H-NMR experiments were all performed on the same day with 32 scans. Samples were prepared with 20 mg of chitosan mixed with 1 mL of D_2_O and 10 µL of DCl (7.4M). The samples were stirred overnight to solubilize the product entirely. The ^1^H-NMR spectra were calibrated from the signal of HOD at 4.79 ppm.

Deacetylation degrees (DA) have been determined using the following Equation (2) [74]:(2)$$\mathrm{DA}=\frac{\left(\frac{1}{3}*I\;CH3\right)}{\left(\frac{1}{6}*I\;H2-6\right)}\;\times \;100$$

The measurement of DPs is based on the comparison of the integration of the proton H_2_ at 3.10 ppm with 1 of the protons of 2,5-anhydro-D-mannose at the reducing end of the chain at 5.35 ppm [60,61]. DPs have been determined using the following Equation (3): (3)$$\mathrm{DP}=\frac{I\;H2}{I\;H2,5-anyhdro-D-Mannose}$$

#### 4.2.3. SEC MALS

Polymer molar masses were determined by Size Exclusion Chromatography (SEC) using an acidic buffer as the eluent. Measurements were performed on an Ultimate 3000 system from Thermoscientific (Waltham, MA, USA) equipped with a diode array detector (DAD), a multi-angle light scattering detector (MALS) and a differential refractive index detector (dRi) from Wyatt technology. Polymers were separated on 2 connected G4000PWXL and G3000PWXL gel columns (300 mm × 7.8 mm) (exclusion limits from 200 Da to 300,000 Da). The column temperature was held at 25 °C. Two buffers were used (sodium acetate 0.2M and acetic acid 0.3M; ammonium acetate 0.15M and acetic acid 0.3M) at a flow rate of 0.6 mL/min. Samples were solubilized at a concentration of 5 mg / mL overnight and filtered with 1.2 µm, 0.45 µm and finally 0.22 µm to remove all insoluble material.

The specific refractive index increment (dn/dc) corresponding to the dependence of the solution’s refractive index on solute concentration was determined. The dn/dc of a polymer depends on the chemical composition of the polymer, the solvent and the temperature used and also the wavelength of the incident laser. The dn/dc of a polymer is usually considered as a constant in a given solvent; however, the contribution of end groups becomes significant at lower molecular weights, leading to variation of dn/dc. Individual offline batch-mode measurements were performed to determine chitosan accurate dn/dc values in ammonium acetate and sodium acetate buffers. Various polymer concentrations were prepared, ranging from 0.5 mg/mL to 5 mg/mL and injected at 0.6 mL /min. Differential refractive index data were obtained for each solution and plotted versus concentration using the Wyatt Astra VII software. The slope of the linear fit is proportional to the dn/dc of the polymer for this particular solvent. The values of dn/dc are given in Appendix A.

#### 4.2.4. Elemental Analysis (C, H, N, O)

Elemental analysis was performed by SGS (Evry-Courcouronnes, France). 

#### 4.2.5. XPS

A ThermoFisher Scientific K-ALPHA spectrometer (Waltham, MA, USA) was used for XPS surface analysis with a monochromatized Al-Kα source (hν = 1486.6 eV) and a 400 μm X-Ray spot size. Powders were pressed onto indium foils. The full spectra (0–1100 eV) were obtained with a constant pass energy of 200 eV, while high-resolution spectra were recorded with a constant pass energy of 40 eV. Charge neutralization was applied during the analysis. High-resolution spectra were quantified using the Avantage software provided by ThermoFisher Scientific. The main attention was paid to the Ca 2p spectra to determine the calcium chemical environment.

#### 4.2.6. FTIR

Fourier transform infrared (FTIR) spectra were recorded on a Bruker VERTEX 70 instrument (Billerica, MA, USA) (4 cm^−1^ resolution, 64 scans, DLaTGS MIR) equipped with a Transmittance bank adapted to KBr pellets. KBr pellet was prepared with 2 mg of the desired product, and 198 mg of KBr previously dried in an oven to prevent water contamination. The 200 mg mixtures were then compressed with a press at 10 tons for 1 min to make the pellets.

#### 4.2.7. Mass Spectrometry

##### Electrospray

Electrospray analyses were performed on a linear trap quadrupole (LTQ) mass spectrometer (Thermo Fisher Scientific, Waltham, MA, USA) in positive ion mode using direct infusion of the samples in a mixture water/methanol (4/1, *v/v*) (0.1 mg/mL). Electrospray source parameters were as follows: capillary voltage +20 V, tube lens voltage +90 V, capillary temperature 300 °C, sheath and auxiliary gas flow (N2) 8 and 5, sweep gas 0, spray voltage 3.6. MS spectra were acquired by full range acquisition covering *m/z* 50–2000. 

##### MALDI-TOF

MALDI-MS spectra were registered on a Voyager mass spectrometer (Applied Biosystems Waltham, MA, USA) equipped with a pulsed nitrogen laser (337 nm) and a time-delayed extracted ion source. Spectra were recorded in the positive-ion mode using the reflectron or the linear mode with an accelerating voltage of 20 kV. The chitosan samples were dissolved in a mixture made of H_2_O/MeOH (50/50 *v/v*) and acetic acid (0.1%, *v/v*) at 10 mg/mL. The 2,5-dihydroxybenzoic acid (DHB) matrix solution was prepared by dissolving DHB (10 mg) in MeOH (1 mL). A MeOH solution of cationization agent (NaI, 10 mg/mL) was also prepared. The solutions were combined in a 10:1:1 volume ratio of matrix to sample to cationization agent. One to two microliters of the obtained solution were deposited onto the sample target and vacuum-dried.

##### Pyrolysis

Py-GC/MS was carried out using a Single-Shot Pyrolyzer (PY-3030S, Frontier Lab Frontier Lab (Fukushima, Japan)) linked to a Thermo ISQ GC/MS system. Chitosan samples (~1 mg) were pyrolyzed at 500 °C. The pyrolysis products were separated on a capillary column (Optima-5-MS, 30 m × 0.25 mm × 0.25 μm). The column temperature was initially held at 50 °C for 1 min and then ramped at 6 °C min^−1^ to 300 °C and held at this temperature for 2 min with a constant helium 6.0 flow rate of 1.2 mL min^−1^. The injection temperature was at 230 °C with split mode. The transfer line and ion source temperatures were maintained at 250 and 200 °C, respectively. The MS was operated in the full scan mode (70 eV), scanning the mass range of 40–800 amu. The components generated from the pyrolysis of chitosans were identified by the software of NIST MS search 2.0 (Gaithersburg, MR, USA).

#### 4.2.8. TGA 

TGA analyses were done on a TA Instruments Q500 (New Castle, DE, USA) under nitrogen from ambient temperature to 800 °C at a rate of 10 °C/min. The flow rate was set to 40 mL/min for the balance and 60 mL/min for the furnace. Gas flow was then changed to air until 950 °C to be able to determine the ash percentage. Analyses were carried out with 15 mg of chitosan per analysis.

#### 4.2.9. *Botrytis cinerea* and Downy Mildew Assays

For *B. cinerea* growth inhibition assays, the direct antifungal activity of chitosan was assessed by growing 270 µL of *B. cinerea* conidia (2.10^5^ c/mL) in Potato Dextrose Broth ¼ diluted with 30 µL of different final concentrations of chitosan (25, 50, 100, 250 and 500 mg/L), in a 100-wells microplate honeycomb Bioscreen. The growth of *B. cinerea* was followed by optical density at 492 nm using the Thermo Labsystem Bioscreen C system (cyan filter) with a reading every 2 h for 60 h (20 °C, dark, continuous agitation). CHI-1 and CHI-2 were dissolved in sterile ultrapure water, and CHI-3, CHI-4 and CHI-5 were dissolved in lactic acid pH4.5 (0.1%, 0.2% and 0.3%, respectively). The half-maximal inhibitory concentration (IC50) was determined. It corresponds to the point of intersection of the growth inhibition curve when the inhibition is 50%. 

For toxicity tests on *P. viticola* zoospores, a suspension of *P. viticola* sporangia (1.10^5^ sp/mL) prepared in osmosed water was treated with different final concentrations of chitosan (0.001, 0.005, 0.01, 0.1, and 1 g/L from a stock solution. An hour and a half later, released zoospores, moving on a 1 mm^2^ square of a Malassez hemocytometer, were counted for one minute. CHI-1 to CHI-5 were prepared as described above in the *B. cinerea* growth inhibition assays.

## 5. Conclusions

Chitosan of lower molecular weight showed very good bioactivity results against *P. viticola* and *B. cinerea* and had a great potential to be used as a biopesticide for vineyards, even if that chitosans are in different forms. Neither chlorine, calcium, nor protonated chitosan negatively impacted the antifungal properties of chitosan. The presence of chlorine or calcium can possibly happen when process productions are not properly adapted. Chitosan under hydrochloride form and with chlorine atoms can happen when neutralization is not properly carried out. In the same way, calcium atoms can be present when chitosan is not properly washed after a bleaching process. Although, it can also happen when a demineralization step is not properly carried out. Depending on the application, the presence of such atoms is not necessarily an inconvenience as chitosan could still be used as a biopeptide in vineyards as it showed good bioactivity against *P. viticola* and *B. cinerea.* Not doing the neutralization step properly could save production costs for industrials. However, for academic purposes, this can be an issue in properly being able to correlate biological properties to physicochemical characteristics. When buying commercial chitosan, purity should always be verified as companies tend to struggle to reach a standardized product due to the differences in the initial biomass composition and from their own process. 

## Figures and Tables

**Figure 1 molecules-28-00966-f001:**
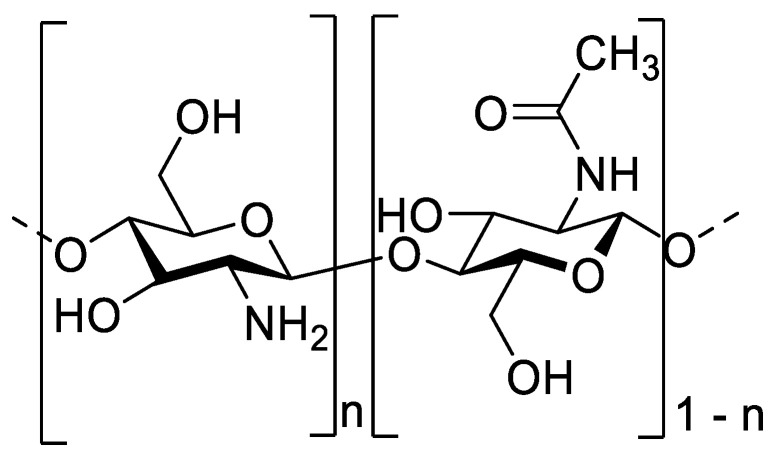
Chemical structure of chitosan with glucosamine (D unit) and acetyl-glucosamine (A unit) units randomly distributed along the chain.

**Figure 2 molecules-28-00966-f002:**
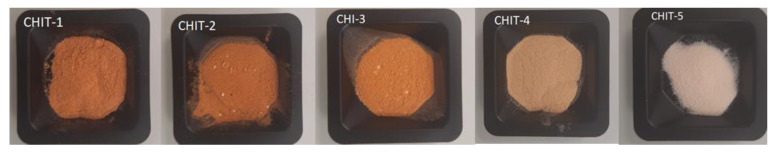
Views of all different chitosans from CHI-1 (**left**) to CHI-5 (**right**).

**Figure 3 molecules-28-00966-f003:**
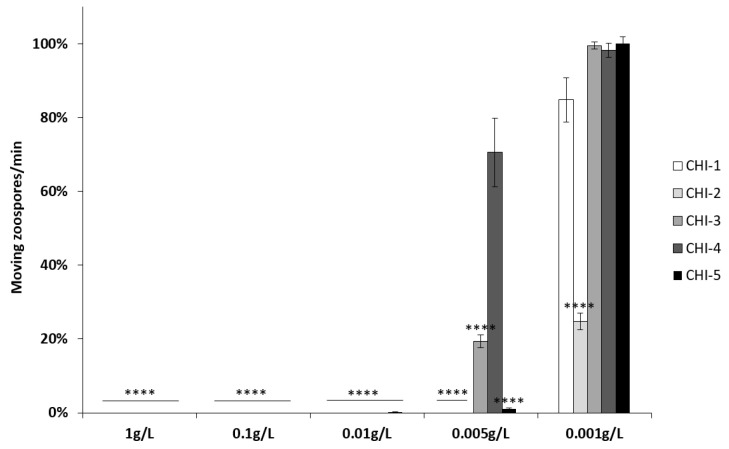
Toxicity effects of CHI-1 to CHI-5 on the motility of *Plasmopara viticola* zoospores. *P. viticola* sporangia were treated with increasing concentrations of CHI-1 to CHI-5, and released zoospores moving on a 1 mm^2^ square of a Malassez hemocytometer were counted for one minute. Values represent the mean ± SE (*n* = 9) of three independent experiments and are expressed as a percentage of the control, set as 100%. Asterisks indicate significant differences relative to the control using an unpaired heteroscedastic Student’s *t*-test; ****, *p* < 0.0001).

**Figure 4 molecules-28-00966-f004:**
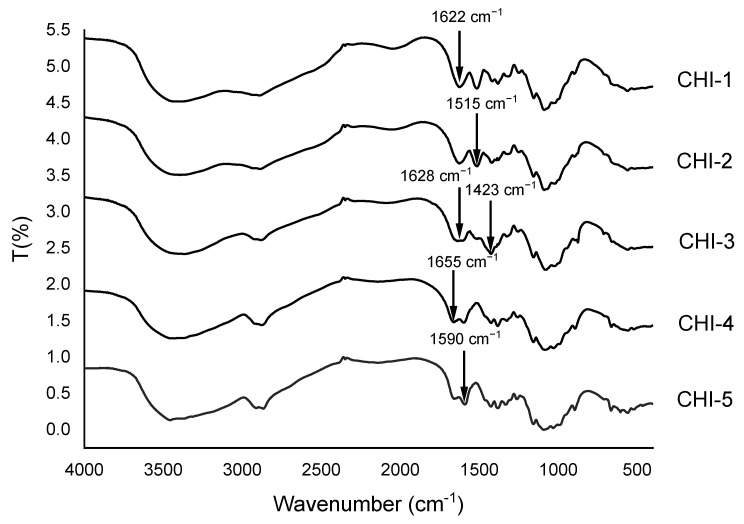
FTIR spectra of the selected chitosans.

**Figure 5 molecules-28-00966-f005:**
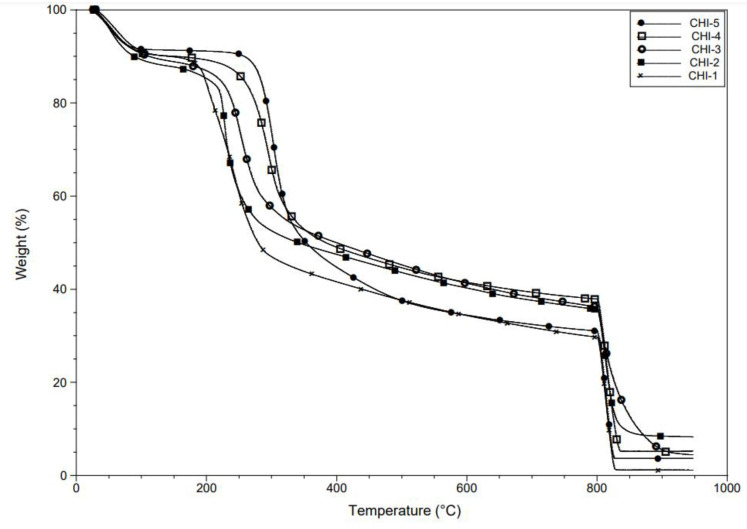
TGA of the selected chitosans under nitrogen atmosphere until 800 °C then air atmosphere until 950 °C.

**Table 1 molecules-28-00966-t001:** Half maximal inhibitory concentration of CHI-1 to CHI-5 on the growth of *Botrytis cinerea* in vitro. *B. cinerea* conidia were treated with increasing concentrations of CHI-1 to CHI-5, and mycelial growth was followed by optical density. The IC50 was determined by measuring the concentration of chitosan required to inhibit the growth of B. cinerea by 50% (Appendix A). Values represent the mean of the IC50 (mg/L) ± standard error (SE) of triplicate data obtained in three independent experiments (*n* = 9).

Sample	IC50 (mg/L)	SE
CHI-1	13	0
CHI-2	13	0
CHI-3	17	8
CHI-4	75	47
CHI-5	152	78

**Table 2 molecules-28-00966-t002:** Antioxidant properties of the different chitosans. The data are given as mean ± standard deviation (*n* = 3).

Sample	Scavenging Effect (%) at 1.6 mg/mL
CHI-1	7.9 ± 1.7
CHI-2	78.9 ± 4.3
CHI-3	75.6 ± ND *
CHI-4	36.1 ± 4.1
CHI-5	5.7 ± 1.9

* Not determined.

**Table 3 molecules-28-00966-t003:** Deacetylation degrees (DAs) of the selected chitosans. The data are given as mean ± standard deviation (*n* = 3).

Sample	DA NMR (%)
CHI-1	98 ± 0.2
CHI-2	98 ± 0.1
CHI-3	95 ± ND *
CHI-4	94 ± ND *
CHI-5	93 ± ND *

* Not determined.

**Table 4 molecules-28-00966-t004:** The molecular weight of selected chitosans from SEC-MALS in NH_4_Ac buffer or NaAc buffer. Data are means using three replicates. The data are given as mean ± standard deviation (*n* = 3).

Buffer	Sample	Mn (kDa)	Mw (kDa)	*Đ*	*DPn*
NH_4_Ac	CHI-1	6.8 ± 0.1	9.3 ± 0.1	1.3 ± 0.1	42.3 ± 0.1
	CHI-2	8.0 ± 0.1	12.1 ± 0.7	1.5 ± 0.1	49.7 ± 0.6
	CHI-3	11.0 ± 1.2	16.1 ± 2.1	1.4 ± 0.1	69.0 ± 7.0
	CHI-4	22.5 ± 0.7	52.9 ± 5.9	2.3 ± 0.2	138.8 ± 4.6
	CHI-5	55.0 ± 8.5	99.2 ± 29.4	2.7 ± 0.3	339.7 ± 53.0
NaAc	CHI-1	10.5 ± 2.2	11.8 ± 0.2	1.1 ± 0.1	64.8 ± 14.1
	CHI-2	14.4 ± 2.1	17.5 ± 2.3	1.2 ± 0.1	89.3 ± 12.9
	CHI-3	17.3 ± 3.5	41.9 ± 2.9	2.5 ± 0.6	106.9 ± 21.7
	CHI-4	36.3 ± 2.2	122.9 ± 23.9	3.4 ± 0.8	224.4 ± 13.61
	CHI-5	69.4 ± 5.2	186.9 ± 36.4	2.7 ± 0.4	428.6 ± 32.6

**Table 5 molecules-28-00966-t005:** Depolymerization degree of CHI-1 to CHI-3 from ^1^H-NMR at room temperature in D_2_0: DCl mixture. The data are given as mean ± standard deviation (*n* = 3).

Sample	DP (NMR) *
CHI-1	15.4 ± 0.7
CHI-2	43.8 ± 2.1
CHI-3	113.0 ± ND **
CHI-4	-
CHI-5	-

* Determined with equation (3). ** Not determined.

**Table 6 molecules-28-00966-t006:** Elemental analysis of the selected chitosans *.

Element (%)	Carbon	Hydrogen	Nitrogen	Oxygen	Total	Unknown
CHI-1	32.2	6.72	6.22	38.86	84.00	≈−16
CHI-2	33.0	6.64	6.09	39.74	85.47	≈−14
CHI-3	34.4	6.32	6.32	41.88	88.92	≈−11
CHI-4	39.8	6.85	7.40	43.86	97.91	≈−2
CHI-5	40.6	7.07	7.57	43.61	98.85	≈−1

* Uncertainty (%): C ± 0.4; H ± 0.2; N ± 0.2; O ± 0.4.

**Table 7 molecules-28-00966-t007:** Mass concentration (%) of surface elements detected by XPS.

Element	CHI-1	CHI-2	CHI-3	CHI-4	CHI-5
Carbon	41.5	42.9	41.0	59.1	57.7
Nitrogen	6.2	6.2	5.4	5.2	5.0
Oxygen	31.3	31.6	33.0	26.8	26.9
Chloride	13.1	9.4	5.2	0.6	0.3
Calcium	0.5	2.8	7.0	1.3	2.8
Silicium	0.7	0.5	2.4	0.2	0.2
*Hydrogen **	6.7	6.6	6.3	6.8	7.1
*Sum of hetero atoms*	14.3	12.7	14.3	2.1	3.3

***** Percentage obtained from elemental analysis.

**Table 8 molecules-28-00966-t008:** DTGmax, water content and ash content determined from TGA analysis on the selected chitosans.

Sample	DTGmax (°C)	Water Content (%)	Ash Content (%)
CHI-1	201.4/241.7	10.0	1.2
CHI-2	227.9	12.0	8.3
CHI-3	251.7	11.0	4.4
CHI-4	294.7	9.9	5.2
CHI-5	302.9	8.6	3.7

## Data Availability

The data presented in this study are available on request from the corresponding author.

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
