# Peer review of "Deep Chemical and Physico-Chemical Characterization of Antifungal Industrial Chitosans—Biocontrol Applications"

_molecules, 2023, doi:10.3390/molecules28030966_

Round 1

Reviewer 1 Report

The authors provided a well-written manuscript, which describes chemical and physico-chemical characterization of different industrial chitosans. The article is well written and the results reported seems correct. Combining different experimental techniques the authors found the correlation between the purity of 5 different chitosans and its antifungal activity. The results look very promising for the development of novel viticultural techniques. 

Author Response

The authors thank the reviewer 1 for his/her comments.

All comments have been taken into consideration as suggested

Reviewer 2 Report

1. Details of 5 samples is missing. What is the variable in in C1-C5.

2. How the different 5 samples prepared and classified. 

3. There is no significant difference in the sample was observed by IR spectroscopy.

4. Why 1.6mg/ml concentration selected for the scavenging activity.

5. How the MIC level calculated. 

Author Response

The authors thank the reviewer 2 for his/her comments. All the suggestions were taken into consideration.

1- Details of 5 samples is missing. What is the variable in C1-C5?

The authors do not understand very well this comment because the paper deals with a deep characterization of each chitosans. The main difference between all the chitosans is their size.

2- How the different 5 samples prepared and classified?

In this work, the industrial samples were provided without any description other than the fact that they were obtained from crustacean shells with a very high degree of deacetylation. The analyses of the degree of polymerization led the authors to a classification.

3-There is no significant difference in the sample was observed by IR spectroscopy

There is a significant difference between the samples, but the authors admit that it is no so easy to observe on the Figure 4. For example, there is one wavenumber attributable to NH2 and another one to NH3+

4- Why 1.6 mg/mL concentration selected for the scavenging activity?

In the litterature, there are different protocols and no well established procedure. The authors used to using these experimental conditions. But no matter which protocol is selected, the important thing is to use the same one to be able to compare the values, or else, you have to take a reference in parallel. We have used the same protocol, so our data are comparable between them.

5- How the MIC level calculated?

The half maximal inhibitory concentration (IC50) corresponds to the point of intersection of the growth inhibition curve when the inhibition is 50%. This explanation was added in the manuscript.